# Enhanced fish production during a period of extreme global warmth

Gregory L. Britten ⬤ [1,5✉] & Elizabeth C. Sibert ⬤ [2,3,4,5]

Marine ecosystem models predict a decline in fish production with anthropogenic ocean warming, but how fish production equilibrates to warming on longer timescales is unclear. We report a positive nonlinear correlation between ocean temperature and pelagic fish production during the extreme global warmth of the Early Paleogene Period (62-46 million years ago [Ma]). Using data-constrained modeling, we find that temperature-driven increases in trophic transfer efficiency (the fraction of production passed up trophic levels) and primary production can account for the observed increase in fish production, while changes in predator-prey interactions cannot. These data provide new insight into upper-trophic-level processes constrained from the geological record, suggesting that long-term warming may support more productive food webs in subtropical pelagic ecosystems.

[1] Program in Atmospheres, Oceans, and Climate, Massachusetts Institute of Technology, Cambridge, MA 02139, USA. [2] Department of Organismic and Evolutionary Biology, Harvard University, Cambridge, MA 02138, USA. [3] Society of Fellows, Harvard University, Cambridge, USA. [4]Present address: Earth and Planetary Sciences, Yale University, New Haven, CT 06520, USA. [5]These authors contributed equally: Gregory L. Britten, Elizabeth C. Sibert. ✉email: gbritten@mit.edu

Fish are a significant protein source for much of the world's population[1] and are the dominant vertebrates in marine ecosystems[2]. As upper-trophic-level consumers, fish biomass, productivity, community composition, and size distribution are impacted by changes in the magnitude and composition of primary production[3–6], which in turn are impacted by changes in climate[4,5]. Earth system models consistently predict that the primary production supporting fish populations will decline with anthropogenic global warming, mainly due to an intensification of low latitude stratification and reductions in nutrient supply[5,7,8]. Subsequent declines are expected in fish production as reductions in primary production are passed through the trophic food web via trophic transfer processes[4,5,9]. Indeed, fisheries simulations and global data compilations have shown significant declines in fish productivity over recent decades in response to anthropogenic ocean warming[10–13], with low latitude subtropical ecosystems (~20–40° latitude) hardest hit due to the negative effect of stratification on primary production[4,5,14].

Despite consistent model-based predictions of declining primary and higher trophic-level production with anthropogenic-scale ocean warming, geological records are less clear regarding the relationship between global ocean temperature and marine productivity on long timescales, with some records showing positive correlations, and others showing negative correlations or no relationship at all[15–19]. Furthermore, the sedimentary record of marine production is limited to the tiny fraction of the ecosystem that makes it to the seafloor[17], primarily the fossils of biomineralizing phytoplankton[20] (calcareous coccolithophores and siliceous diatoms) which make up less than half of modern primary production[21]. Thus quantitative information on primary and higher trophic level production is largely absent from the geologic record, leaving a gap in our understanding of how fixed carbon may transfer up the food web during periods of long-term global warming.

While predictions about the productivity of fish and other higher trophic level organisms often follow from changes in primary production[4,5,14], it is also possible for higher trophic level production to be impacted by changes in the efficiency of food web processes[9,22,23]. Physiological arguments suggest that metabolic costs increase at higher temperature[24,25] which would reduce secondary production due to a greater fraction of energy devoted to basal metabolism[24]. Other predicted impacts of higher temperatures are reduced body sizes due to a reduction in dissolved oxygen[26] and increased metabolic costs[27] which would

further reduce trophic energy transfer. However, comparisons of tropical and temperate oceans show that the fraction of available prey consumed can increase with warming[28,29] which could have a positive influence on trophic efficiency. These observations have been supported in experimental and comparative studies where predator attack rate and foraging success often increase with temperature[30–32]. Niche models built on global phytoplankton datasets further suggest a net positive response of picophytoplankton communities to increasing temperature, particularly in subtropical pelagic ecosystems[33]. Taken together, evidence suggests that increasing temperature shifts a suite of processes related to nutrient supply, grazing, and individual physiology, with the potential for both positive and negative impacts on higher trophic levels.

Here we examine the relationship between fish productivity, temperature, and ecosystem processes over geological time, using a unique record of fish fossil accumulation on the seafloor and a simple trophic transer model. We find strong statistical support for enhanced fish production during a period of extreme global warmth, which our model explains in terms of increased trophic transfer efficiency and primary production.

## Results and discussion

**Ichthyolith accumulation rate**. The Early Eocene Climate Optimum (EECO) occurred ~52–48 Ma and was the warmest Earth has been in the past 100 million years, with reconstructed atmospheric $CO_2$ in excess of 1000–1500 ppm, ocean bottom water temperatures above 10 °C, and sea surface temperatures in excess of 35 °C in the tropics[34,35]. This contrasts with modern anthropogenic emissions where atmospheric $CO_2$ may increase from 385 ppm to well over 1500 ppm in a matter of centuries under the IPCC RCP 8.5 "worst-case" emissions scenario[36]. Despite extreme greenhouse conditions, we find evidence from Deep Sea Drilling Project (DSDP) Site 596, a sediment core from the South Pacific gyre, that the peak warmth of the EECO was associated with a nearly 10-fold increase in subtropical pelagic fish production, recorded by the accumulation rate of microfossil fish teeth and scales (ichthyoliths) in the sediments (ichthyolith accumulation rate [IAR] = ichthyoliths/cm²/million years; see refs. [37–39]; Fig. 1). Ichthyoliths are composed of calcium phosphate, extremely resistant to dissolution, and preserved in excellent condition in the corrosive environment of the deep-sea —in this core going back >85 million years[40]. We find that observed IAR is strongly correlated with an independent

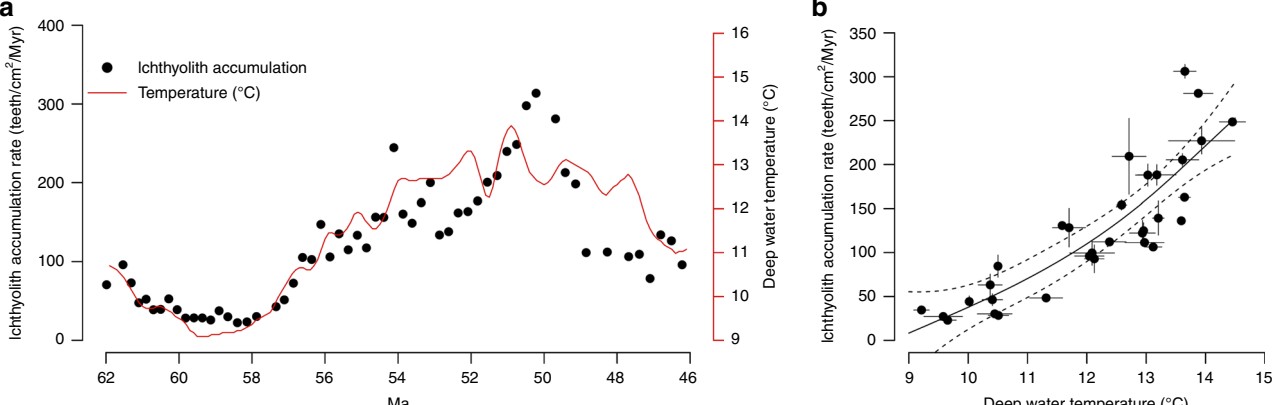

**Fig. 1 Correlation between fish production and temperature 62–46 Ma. a** Observed total ichthyolith accumulation rate (black dots) and paleoceanographic temperature reconstruction derived from δ18O (red line; ref. [41]). **b** Nonlinear regression between total ichthyolith accumulation rate and paleoceanographic temperature, binned by half Myr time intervals. Black dots give the half Myr-binned means for total ichthyolith accumulation rate and temperature, solid lines around the dots give the sample standard deviation. The solid black curve gives the mean regression and dashed black curves give the 95% confidence interval for the regression (n = 32). Source data for (**b**) are provided as a Source Data file.

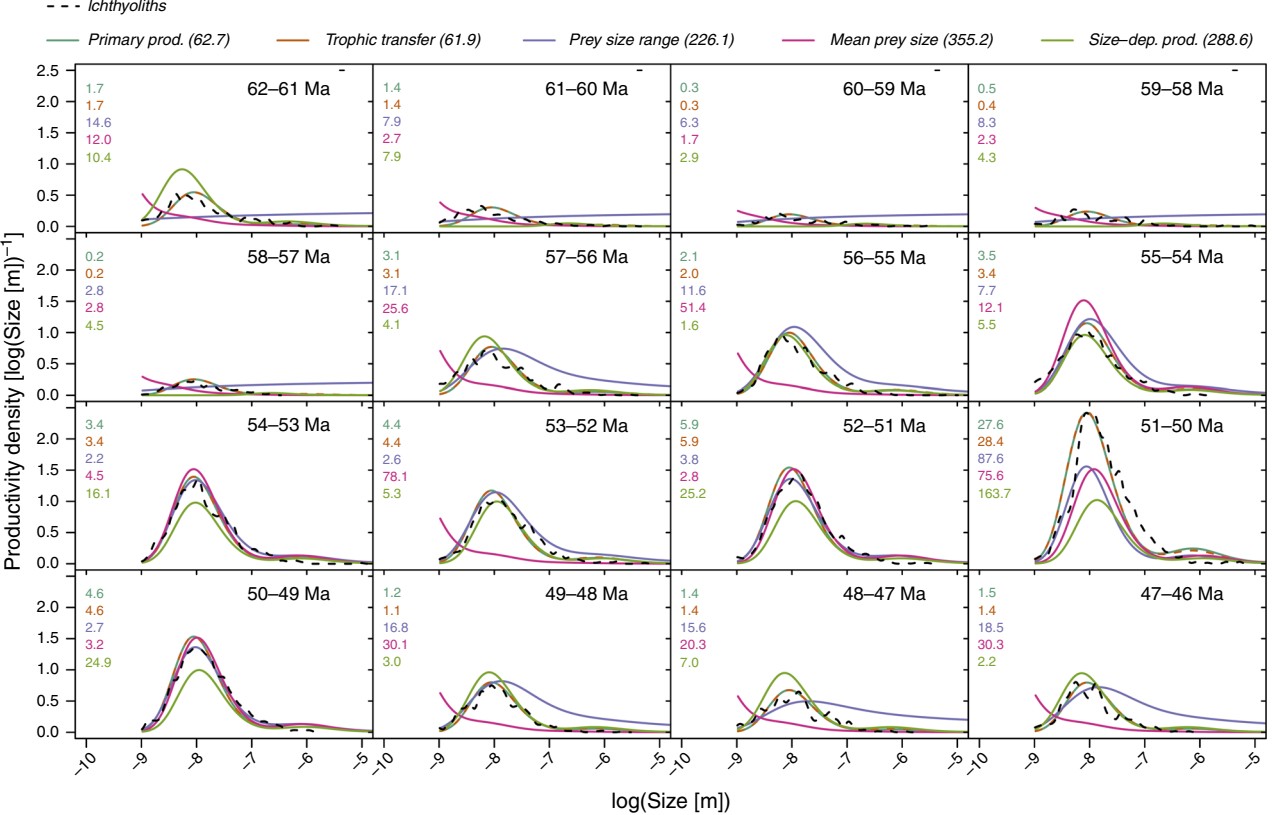

**Fig. 2 Time series and model fits for the ichthyolith size distributions.** Ichthyolith observations (dashed black line) are binned by Myr increments. Model fits are shown for the time-varying primary production model with constant size-productivity scaling (blue), trophic transfer efficiency (orange; shown dashed in the figure due to overlap with the primary production model), prey size range (purple), mean prey size (red), and time-varying primary production with size-dependent scaling (green). The RMSE computed for each Myr bin is displayed in the upper left corner with the same color code and order. The total RMSE across time is given in the top legend in brackets.

$\delta^{18}O$-reconstructed ocean bottom water temperature record[41] throughout the 16 million year interval spanning 62–46 Ma (Fig. 1), suggesting that ocean warming on long timescales was favorable to fish production in this region. We used bottom water temperature rather than sea surface temperature, as the majority of fish biomass in the modern ocean lives below the thermocline distributed throughout the mesopelagic[42], and are likely decoupled from variations in sea surface temperature, but sensitive to the overall variations in global climate across depths captured by the broader bottom-water temperature record.

At DSDP Site 596, total fish community productivity (inferred from total IAR) varies over the period 62–46 Ma, from a low of 30 ich/cm²/Myr at ~60 Ma, to a high near 300 ich/cm²/Myr during the height of the EECO at ~50 Ma (Fig. 1a). A quadratic regression between IAR and reconstructed paleotemperature[41] explains 75% of the variation in total fish productivity (Fig. 1b), and the quadratic term is significant when compared to a linear regression, indicating a positive, non-linear relationship between fish productivity and temperature through the study interval (F-test $p < 0.05$).

**Data-constrained modeling.** To investigate the ecological processes that may underlie the relationship between fish productivity and temperature, we fit an idealized food web model to the time-varying size-abundance distribution of fossil ichthyoliths at DSDP Site 596. The model propagates a size-specific primary productivity distribution through secondary, tertiary, quaternary, and quinary productivity, assuming means and standard deviations in the predator–prey size ratios and fractional trophic transfer efficiencies. While highly idealized, the model is a minimal representation of the size-structured food web that

projects changes in basic ecological processes underpinning marine food webs (primary productivity, predator–prey size ratios, and trophic transfer efficiency) to a predicted size distribution of upper-trophic level productivity which can be related to the preserved fish tooth size-abundance distribution when assuming allometric scaling between tooth and body size[43]. Changing parameters in the model modulates how biomass is passed from phytoplankton to fish, which in turn modulates the size and shape of the upper trophic productivity distribution. Increases in trophic efficiency and primary productivity scale total productivity up and down while leaving the shape and location of the distribution unchanged; changes in mean predator–prey size ratios stretch and contract the distribution along the size axis, while changes in the standard deviation of the predator–prey size smooth out the shape of the distribution (Supplementary Fig. 1). Importantly, we were limited in model complexity by the trophically aggregated ichthyolith record. More sophisticated models could involve trophically defined relationships (e.g., prey size or trophic efficiency increasing/decreasing with trophic level) but cannot be reliably fit to the data due to trade-offs in how they impact the trophically aggregated model output. Using a least-squares approach, we estimate the time-constant model parameters that best fit the observed time series of size-specific fish productivities. We then perform systematic statistical experiments where parameters of the model are allowed to vary through time to capture the time-varying characteristics of the ichthyolith data (see "Methods" for full description of the model and data analysis).

While total fish productivity varied markedly throughout the 16 Myr study interval from 62–46 Ma, the mean and standard

deviation of the ichthyolith size distribution did not (Fig. 2, Supplementary Fig. 2). The mean and standard deviation of tooth length varied by ~5 and 20% (in arithmetic space), respectively, but showed no significant trend over time, nor any significant relationship with temperature. We note the only major diversification of fish tooth morphotypes occurred at the beginning of the record (62–58 Ma; ref. [44]), a time with relatively little change in temperature. There is also no structural or evolutionary change in fish diversity at DSDP Site 596 during the remainder of the study interval (58–46 Ma)[44], suggesting that the observed variation in the ichthyolith record is most likely due to changes in fish production, rather than changes in fish community composition or diversity.

Using the data-constrained model, we find that temporal changes in trophic transfer efficiency most parsimoniously explain the observed ichthyolith time series, followed closely by changes in total primary productivity. Changes in predator–prey size ratios and size-specific primary productivity were less able to reproduce the observations (Fig. 2). These results separate the models into two classes—one class of mechanisms related to total energy flow that captures the observed changes in height of the distribution but does not alter the spread (trophic transfer efficiency and primary productivity), versus a second class of predator–prey mechanisms that modify the structural connections between trophic levels (mean prey size and generalism), and primarily change the spread and smoothness of the distribution. We find that models driving increases in energy flow are able to explain the ichthyolith observations while models driving changes in food web structure cannot. Specifically, including time-variation in trophic transfer efficiency reduces the model root mean squared error (RMSE) by a factor of 13.7 relative to the time-constant reference parameters. Time-varying primary production reduced the RMSE by a similar factor of 13.6. Time-varying mean predator–prey size ratios and predator–prey size ratio standard deviations reduced the RMSE by factors of 2.4 and 3.7, respectively. Comparing time-varying trophic transfer efficiency and productivity, we found that time-varying primary productivity requires up to 100% variation per Myr to fit the data, while trophic transfer efficiency requires only 10% variation per Myr, meaning that relatively small shifts in trophic transfer efficiency have a relatively large impact on fish production.

Our simplest model configuration holds the shape of the primary productivity size distribution constant with increasing magnitude of primary production; however mean cell size of the phytoplankton community is thought to increase with the level of total primary production[45–47]. Including positive linear size-primary productivity scaling in the model increased the optimized RMSE by a factor of three relative to the time-varying model without size-productivity scaling, as increases in the phytoplankton size distribution lead to predictions of larger fish which is inconsistent with the ichthyolith record. Taken together, these extended analyses suggest trophic transfer efficiency as a more parsimonious explanation of the observed ichthyolith record, although changes in primary production should also be considered.

While changes in primary production could be caused by changes in available resources (nutrients, light, temperature), it is less clear what mechanisms might drive temperature-dependent trophic transfer efficiency over geological time. Modern observations in marine ecosystems suggest an increase in grazing rates at warmer temperatures[28,29], largely ascribed to increases in predator–prey attack rates. However we note this would require grazing rates to simultaneously overcompensate for increases in metabolic cost at elevated temperature[24,25]. Increased temperature may also drive increased productivity of the ocean's microbial loop which makes dissolved organics available to

higher trophic levels[6] and may additionally increase trophic transfer efficiency.

Importantly, fish productivity likely responds differently to warming on anthropogenic vs. geological timescales. The anthropogenic timescale is a rapid perturbation of the established ecosystem that occurs over tens of generations, while the multi-million-year warming of the EECO occurred over hundreds of thousands of generations. This suggests that slow and relatively non-perturbative warming may enhance biomass transfer within an established ecosystem without the chaotic reorganization and extinction associated with rapid environmental change[4,43].

The strong numerical correlation between ichthyolith-inferred fish production and an independent global paleo-temperature reconstruction points to a positive nonlinear temperature response of total fish production to ocean warming on million-year timescales—standing in stark contrast to the current paradigm predicting reduced fish productivity with shorter-term anthropogenic warming. While many processes are likely to be acting simultaneously on the observed record, our modeling suggests that trophic transfer efficiency and primary production may be drivers of elevated fish productivity in the subtropical South Pacific during the EECO. With analytical methods now in place to measure the ichthyolith and microfossil record in detail[37], combined here with ecological modeling, this study points in an exciting direction to better understand the role of long-term climate in controlling fish community productivity, synthesizing ecological models with evidence from the fossil record over periods of extreme global change.

## Methods

**Data**. Isolated fish teeth (ichthyoliths), were recovered from DSDP Site 596, a red-clay sediment core from the center of the South Pacific Gyre. Samples were washed over a 38 μm sieve, and all ichthyoliths >106 μm were picked out of the sediment and mounted on cardboard microfossil slides. The samples were then imaged using a microscope-mounted Canon Powershot S5 IS camera. The length of each recovered ichthyolith was measured using ImageJ, and each image was size-calibrated using an ocular micrometer (the data analyzed were originally gathered and published in ref. [43]). At each time point the records were converted to a total IAR (units: # teeth/cm$^2$/Myr) using an established age model for the site[48]. The study interval of 62–46 Ma was chosen to begin after the rapid evolution of fish following the Cretaceous-Paleogene Mass Extinction, as well as to avoid the potential influence of ice volume on δ$^{18}$O-derived temperature estimates into the middle Eocene. To provide smoothing and reduce noise in the time-varying parameter model analyses, we grouped the ichthyolith data into 1 Myr bins. Each bin contained four measurements, on average, with an intra-sample standard deviation of 13.8%. The final analyzed dataset consisted of a per-Myr time-varying tooth-size distribution where the area under the curve (the total IAR) is a proxy for total fish community production. We adopt this simple measure of total relative fish community production following the previous work[38,43,44].

**Trophic model**. We modeled the time-varying tooth size distribution by first assuming an allometric scaling relationship between tooth size and body size of the form

$$S_{body} \sim a S_{tooth}^b, \qquad (1)$$

where we assume the parameters $a$ and $b$ are unknown parameters to be calibrated with the ichthyolith record. We note that power-law relationships between tooth size and body size are regularly observed in fish[49,50] but allometric coefficients vary widely, so the estimated parameters are assumed to represent a community average. We divide the food web into $N$ size class bins logarithmically spaced from $10^{-7}$ to $10^1$ [m]. The total productivity at a particular size class $i$ in a particular trophic level is given by the sum over all the size classes in the trophic level below, weighted by the proportion of that size class in the diet and multiplied by the trophic transfer efficiency; specifically

$$P_{i,l} = \alpha \sum_{i=1}^{N} \gamma_{i,l-1} P_{i,l-1}, \qquad (2)$$

where α is the trophic efficiency, $\gamma_{i,l-1}$ is the proportion of size $i$ at trophic level $l-1$ in the diet of size $i$ at trophic level $l$. In this way the model follows basic trophic pyramid principles like that assumed in previous trophic models[9,51]. We chose this framework as a minimal model that can represent size-dependent flows in the food-web. The size-specific diet proportions follow a Gaussian kernel across

size classes[52] of the form

$$\gamma_{i,l-1} = \frac{1}{\sqrt{2\pi\sigma^2}} e^{-\frac{(i-\phi i)^2}{2\sigma^2}}, \quad (3)$$

where $\phi$ and $\sigma$ are parameters that set the mean and variance of the predator–prey size ratios; $\phi$ sets the most preferred prey size for predator of size $i$ in terms of a fraction of the predator size, while $\sigma$ determines the proportional spread around the preferred prey size. Primary productivity at the base of the food web is described proportional to a Gaussian distribution

$$P_{i,1} \propto e^{-\frac{\left(P_1^\mu-i\right)^2}{2\left(P_1^\sigma\right)^2}}, \quad (4)$$

where $P_1^\mu$ is the mean of the primary production size distribution and $P_1^\sigma$ is the standard deviation. The proportionality is due to the unknown conversion between tooth number and biomass and also the unknown proportion of fish in the larger predatory community which may include, for example, cephalopods, gelatinous mesopelagic organisms, and other taxa not represented in the ichthyolith record. To account for this, we scaled the productivity size distribution such that the reference solution has total productivity of one unit per Myr and changes in productivity over time can be modeled relative to the reference solution in %. This procedure assumes that the unknown tooth number per biomass conversion remains constant over time, a reasonable assumption in this case as there are no major changes in fish morphotype community composition that would indicate a major shift in the fish community or biology[44]. We note that a power-law distribution was tested for the primary productivity distribution but we could not find a satisfactory model solution. We also note that the estimated allometry could be used to extrapolate total IAR to total biomass production; however, the lack of trends in the ichthyolith size distribution would not alter the correlation between productivity and temperature.

**Model fitting**. We first calibrate a reference solution for the model by assuming fixed parameters over the full ichthyolith record. The misfit between the model and data are computed as

$$SS = \sum_i \left(f_i - \hat{f}_i\right)^2, \quad (5)$$

where $f_i$ is the observed (kernel-smoothed) tooth density in size bin $i$ and $\hat{f}_i$ is the model-predicted tooth density in size bin $i$. We found the static reference parameters $\theta = \{a, b, \alpha, \sigma, \phi, P\mu, P\sigma\}$ that minimized the sum of squares, yielding $\hat{a} = 0.095$, $\hat{b} = 0.80$, $\hat{\sigma} = 0.236$, $\hat{\phi} = 0.096$, $P_\mu = 50.01\,\mu m$, $P_\sigma = 5.02\,\mu m$. While 50 μm is a very large cell size for a subtropical gyre relative to the modern ocean, we note that this parameter trades off with the allometric parameters in the optimization and so we cannot confidently infer the true underlying phytoplankton size distribution from fish teeth alone. We interpret the parameters primarily as reference values to quantify the impact of positive and negative variations. We then evaluated whether changes in these parameters are able to account for the temporal dynamics seen in the dataset. We binned the time series into 1-Myr bins and estimated a time-varying parameter in each Myr bin by optimizing the sum of squares independently in each bin. We did this for each individual parameter while holding the others at their reference values. We were unable to perform multi-parameter time-varying analysis due to under-determination. As a sensitivity, we also performed the time-varying analysis by constraining the interannual time-variation in the parameters by varying degrees. We applied the constraints that the time-varying parameter could only vary by 5, 10, 20, 30, 40, 50, and 100% per Myr bin. As an additional sensitivity, we also implemented a model that assumes $P_\mu$ scales linearly and positively with total changes in primary productivity as widely observed[45–47]. As explained in the main text, this model does not fit the data well, as increases in $P_\mu$ directly increase the ichthyolith size distribution and is inconsistent with the ichthyolith data. Results of the time-varying parameter optimization are given in Supplementary Fig. 3. All calculations were carried out in R version 3.6.1.

**Reporting summary**. Further information on research design is available in the Nature Research Reporting Summary linked to this article.

## Data availability
The ichthyolith accumulation data analyzed in this study are available at https://doi.pangaea.de/10.1594/PANGAEA.846789. The individual ichthyolith size measurements are available at https://doi.org/10.5281/zenodo.4095198. The raw and imageJ-processed images used in this study are available at https://doi.org/10.5061/dryad.prr4xgxj4. Source data are provided with this paper.

## Code availability
The code developed for this manuscript is available at https://doi.org/10.5281/zenodo.4095198.

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

## Acknowledgements

The authors thank the International Ocean Discovery Program (IODP) and its predecessors for the collection and curation of samples. This work was supported by the Simons Foundation Postdoctoral Fellowship in Marine Microbial Ecology (G.L.B.) and the Harvard Society of Fellows (E.C.S.).

## Author contributions

E.C.S. and G.L.B. conceived the study. Ichthyoliths were isolated and processed by E.C.S. The trophic model was developed by G.L.B. with input from E.C.S. Both authors contributed equally to the project and manuscript, and first-authorship was determined by a coin toss.

## Competing interests

The authors declare no competing interests.
