## [Peer Review File · Nature Communications]

Reviewers' Comments:

Reviewer #1:

Remarks to the Author:

This is interestingly controversial manuscript (I am saying positively).

Since oligotrophication is predicted with global warming, we tend to think fish production would decrease with warming. The authors found the opposite in paleo record in a much longer time scale. The data and result look robust.

It's very important to use paleo-records to understand the present and future ecosystem in proper context, because paleo-records tend to be the only long-term biotic records beyond a few decades. This manuscript does a great job in this sense.

The text is well written in general and I enjoyed the reading a lot.

A few moderate comments are:

1. Fig 1 says Antarctic Bottom Water Temperature, but this isn't based on global 180 stack, so global deep-water temperature? Sorry if I am wrong.

2. It's interesting the fish production has steeper/higher peak at 50 Ma and then rapidly decrease compared to the deep-water temperature (Fig1)

This fish pattern is more similar to SST than deep-water temperature, as the authors can see in Norris et al 2013 Science Fig2 (<https://science.sciencemag.org/content/341/6145/492>). I am not sure if SST data is reasonably available to use for the modeling here. But at least it's worth discussing.

3. Similarly where are fishes from in water column? I mean fish teeth and scales from entire water column deposit on the sea floor eventually.

The authors compare the fish result with bottom water temperature that means they are mainly from deep-sea fishes?

But fish production must be much higher in ocean surface or shallow marine zone. A bit related to my comment above, so it reflects surface/shallow-water biomass and so makes more sense to compare with SST?

Anyway, I think this is an important topic to discuss in the manuscript.

4. Recent papers showed that pelagic diversity-temperature relationship is unimodal (eg Yasuhara et al 2020 PNAS <https://www.pnas.org/content/early/2020/05/20/1916923117>). Similarly it may be possible that productivity relationship is negative in certain temperature range and positive in in certain temperature range?

Also, Yasuhara et al showed that diversity is highest in the tropical edge/subtropics (ie bimodal latitudinal diversity gradient) in warmer worlds. This study showed biomass is also high there in warmer times. This may mean fishes escaped from too hot tropics to tropical edge/subtropics, resulting high diversity and biomass there in warmer worlds like Eocene and future with RCP 8.5. I think it's interesting to add a discussion on this.

Specific comments:

It's good that the abstract explain what is "trophic transfer efficiency" a bit more.

"subtropical" means Paleogene one or present-day one? Or it means more climate regime rather than latitude?

It may be good to better clarify.

Line 22 and 24, good to add Mora et al 2013 PlosBiology doi:10.1371/journal.pbio.1001682 to "4,5"

and/or "5,7"

Line 34 "the tiny fraction of the ecosystem that makes it to the seafloor": unclear a bit. rephrase?

Line 35, Benthic foraminiferan accumulation rate is widely used proxy for surface primary production. The authors may a bit mention/discuss this with citing Yasuhara et al 2012 Paleobiology (<https://doi.org/10.1017/S0094837300000464>); Herguera 2000 MarMic ([https://doi.org/10.1016/S0377-8398\(00\)00041-4](https://doi.org/10.1016/S0377-8398(00)00041-4)); Thomas et al 1995 Paleocyanography (<https://doi.org/10.1029/94PA03056>); etc.

Also, there are organic carbon and calcium carbonate based estimation of primary production widely used.

See eg Yasuhara et al 2009 PNAS (<https://doi.org/10.1073/pnas.0910935106>) and references therein.

Line 61 Subtropical (climate? latitudinal position?) at that time?

Line 64 Good to briefly mention the method of temperature reconstruction (eg delta 18O based).

Line 111 etc Myr or myr. Good to be consistent.

Line 138-140: The last sentence. The authors may like to broaden the sentence to eg usefulness of microfossil and ocean drilling for ecological and evolutionary research in general beyond the fish with citing recent "biotic response" reviews of Norris et al 2013 Science and Yasuhara et al 2017 Biological Reviews (<https://onlinelibrary.wiley.com/doi/abs/10.1111/brv.12223>). It may help to reach wider audience.

I hope this helps to improve the manuscript.

Please do not hesitate to contact me if there is any unclear point at moriakiyasuhara@gmail.com or yasuhara@hku.hk.

Sincerely yours,
Moriaki Yasuhara

—End of the report—

Reviewer #2:

Remarks to the Author:

Review of Britten and Sibert for NatComm

Britten and Sibert assess potential causes for observed changes in the size distribution and abundance of fish debris in the Pacific subtropics over the time window encompassing the early Eocene climatic optimum. They note a strong positive correspondence between inferred fish production based on ichthyolith accumulation rate (IAR) and paleotemperature, and model results suggest that changes in trophic efficiency can best explain observed variation in the size distribution over time.

The question of how global climate change affects marine primary production and overall ecosystem production is longstanding and as-yet unresolved. Studies on short-term time scales suggest a negative impact of temperature, while deep-time records hint at the opposite relationship, or at best a complex one. Britten and Sibert use data on fossil fish teeth and debris to estimate changes in

production at higher trophic levels and demonstrate a remarkable concurrence between IAR and paleotemperature over the most significant interval of climate warming in the last 65 million years. Figure 1 alone is so surprisingly good that it almost doesn't matter what else the authors say in the text. On a geological time scale, the warmer it gets, the more fish debris there is, in a setting that is otherwise fairly constant in paleoenvironment. This result in itself is enough to warrant publication, as it offers clear solid evidence for higher fish production with warming, contrary to expectations of anthropogenic impacts and pointing to a different set of ecosystem responses to the same perturbation on different time scales.

With respect to this result, the only thing I'd like to see mention of, and perhaps it was addressed more explicitly in earlier work that I do not have access to at the moment, is what kinds of taphonomic and depositional processes might give rise to variation in IAR, and could these potentially be responsible for the observed pattern. For example, do fish teeth typically survive passage through the gut of a predator? Could more teeth just mean fewer large predators, so more teeth survive? Given that the shape of the abundance distribution doesn't change, only the height, I don't think this is likely. How long does it take to dissolve them on the sea floor? Being phosphate, I suspect this is not a big issue either. It would be nice though to know something about how IAR scales with production over the modern ocean to be sure that the signal is indeed one related to production and not some other process.

Following this, the authors go on to use the size frequency distributions from different time slices to test

ecosystem models that explore the influence of various potentially causal processes. If tooth size scales with body size, then those data can be used to infer trophic structure at levels above the primary producers. A food web model then attempts to reproduce the changes seen over time by changing one of several variables to see how well it captures the observed variation. Because the mean and range of the size distribution do not shift, but overall abundance does, efficiency of energy transfer between trophic levels emerges as the most likely driver over time. Overall production could effect the same change but would likely change the size distribution as well, adding a larger tail, and this is not observed. Predator-prey size ratios and/or variability would also change the mean and range of the distribution and so can be ruled out.

These inferences are quite interesting because one need not change overall primary production to see a change in production at higher trophic levels. This has been a matter of great debate in the literature regarding the fossil record – does primary production change, and if so, how is that change brought about and sustained on a global and/or long-term basis? Perhaps that is not needed at all to explain some of the first-order trends seen in the fossil record. The flip side of this, if primary production is not driving change at higher trophic levels, is how does one change trophic efficiency? This is a discussion that the authors could stand to bolster in their manuscript. Its treatment now is somewhat cursory. Do fish simply eat more of what is available to them? How and why? The authors admit that the mechanism is unclear, but more discussion than what is present in the paragraph beginning on line 121 is really needed.

Lastly, I completely agree that time scale is everything when comparing these results to the response to anthropogenic warming. The former deals with a macroevolutionary response, while the latter must be almost exclusively ecological. One wonders then whether these data really do provide any insights to 'better understand climate impacts on fish communities' in the context of the present (which is how many will read 'climate impacts' when the word 'anthropogenic' is also in the same paragraph). I might tweak the language in that last paragraph to be careful not to conflate the two processes and risk being misquoted in the future.

This is an interesting and compelling dataset and a thought-provoking analysis. The issues dealt with are important and longstanding ones. The model results might raise more questions than answers, but nonetheless seem to be able to rule out some proposed mechanisms for change. The text is well written and concise. This should be published.

Some minor editorial comments:

I was a bit confused in Figure 2 by the colored lines – the influence of primary production and trophic transfer appear to be exactly the same in each of the model runs. If this is not an error, it should be clarified. I was also unclear how the curve for mean prey size was so different in boxes that otherwise appear quite similar (e.g., 53-52 Myr versus those adjacent).

There are some places where a word appears to be missing in the text:

Line 88 ...predatory-prey size ratio? smooths...

Line 116 ...factor of three relative TO the time-varying

Line 118 ... inconsistent with THE ichthyolith record

Line 123-124 ... overcompensate FOR increases in...

Line 274 ...tooth SIZE distribution...?

Line 310 ... increases in Pu... and ARE inconsistent with the ichthyolith data.

Line 316 ...solid lines give the sample SD - do you mean dashed lines?

Line 323 RMSE computed FOR each Myr bin...

Linda Ivany

Reviewer #3:

Remarks to the Author:

The manuscript by Britten and Siebert discusses a multi-million year record of microfossil fish teeth (ichthyoliths) from the South Pacific subtropical gyre, corresponding to a period of extreme global warming during the Early Paleogene Period (62-46 million years ago), showing that fish production (inferred from the total ichthyoliths accumulation rate) is positively correlated with proxies of ocean temperature. The Authors further use a simple model of the marine food web to argue that this correlation is best explained by changes in the efficiency with which energy (or biomass) is transferred from the base of the food web to fish.

This is a concise paper that merges paleo-ecosystem records and modeling to probe the influence of a fundamental environmental variable, temperature, on a fundamental ecological property — fish production. The approach and results are very original and provocative, and if they could hold up to scrutiny they would be very significant for a broad community.

I have no particular reservation on the ichthyoliths record and its correlation with paleo-proxies of temperature, although this is not my field of expertise. In themselves, the paleo-fish results are exciting, and worth of publication. I am more skeptical about the model and ecological interpretation of the ichthyolith record, which form the backbone of the manuscript. While I am supportive of the idea of combining ecological models and paleo-data, I do not think the analysis convincingly support the interpretation of the paleo-record. Additionally, while I appreciate the conciseness of the manuscript, I feel more could be done to support the results and interpretation — with only two figures, very little material is presented to support many of the claims, making it often hard to judge the robustness of the conclusions. Thus, while provocative, I would place low confidence on the idea

that higher temperature drives an increase in fish productivity by way of an increase in trophic transfer efficiency. Given the potentially important implications, this conclusion should be better supported.

General comments

I am skeptical of the interpretation of the total ichthyoliths accumulation rate as a straight proxy for fish productivity, as discussed in the methods (line 272), and implied by Figure 1. This is a technical point, but it is relevant to the results, and also relates to the size-dependent model of fish production and its interpretation. The total number of ichthyoliths is the sum of the ichthyoliths across all sizes sampled. Each ichthyolith size corresponds roughly to a fish size, e.g. following the relationship given by the equation after line 275. Individual fish production (e.g. biomass produced per unit time per fish) is also a function of size, presumably proportional to $\text{mass}^{2/3}$ or $\text{mass}^{3/4}$ (see for example Brown's metabolic theory of ecology, Brown et al., 2004, Ecology). Given that a fish's mass is proportional to size^3 , fish production should be proportional to size^2 or $\text{size}^{2.25}$, depending on the allometric relationship used for fish production. Thus, to estimate the total fish production based on a size spectrum of fish numbers (like the one presented by the Authors), one should multiply the number of ichthyoliths in a given size bin to a term proportional to $(\text{ichthyolith size})^{(b*2)}$ or $(\text{ichthyolith size})^{(b*2.25)}$, before summing up (i.e. integrating) over the ichthyolith size range. In a sense, larger ichthyoliths should be weighted more than smaller ones, because they correspond to larger fish with a proportionally larger productivity. Thus, the assertion in the methods (line 272) that "the area under the curve is a proxy for total fish community production" seems incorrect, or at least incomplete. Given the observations in Fig. 2, it is not clear how the larger weight that should be given to larger fish could compensate for the lower abundance, thus altering the results of the paper, in terms of production by the entire fish community, rather than total number of ichthyoliths accumulated.

I am also worried that the ecological interpretation is strongly dependent on model assumptions that are not clearly laid out or justified, and whose implications are not systematically evaluated and presented to the reader. The model in itself seems simple, but many of its aspects remain unclear, and very few references are provided to justify its formulation. With plenty of space available to present more analyses in the manuscript, the Authors could have strived to show that many of the uncertain model assumption are indeed not relevant for the final conclusions.

For example, is the model supposed to represent just the fish community or the entire predator community? At some sizes, fish may overlap with other predators (zooplankton, crustaceans at small sizes, cephalopods at larger sizes), but it is not clear that this is accounted for in the model, at least considering the roughly 50-fold size range shown in Fig. 2. The assumption seems that within the size range implied by the figure, fish are the only important predators; this may be true or not for the Paleogene ocean, but it should be clearly stated and supported.

The main equation of the model (following line 282) asserts that the production rate in one trophic level and size class is a weighted sum over the productions of all preys at the previous trophic level (with a log-normal prey preference weight). I have a hard time convincing myself that this formulation is correct, given that it is quite different from other size-structured food web models that I have seen in the past (e.g. starting with the archetypal size-structured food web model of Benoit and Rochet, 2004, Journal of Theoretical Biology), where the productivity at one trophic level does not come directly from the prey productivities at the previous trophic level, but from the prey biomass at the previous trophic level, multiplied by some grazing rate kernel. It may be that the formulations are equivalent in the end, but a more solid justification for this equations relating productivities to productivities seems required, given how central it is.

The size spectrum of primary production is assumed to follow a log-normal distribution (given that the size bins are logarithmically spaced), but no justification is provided for this assumption, which in turn probably strongly affects the distribution of the resulting fish productivity spectra in Fig. 2. In general, size spectra in the ocean are better described by power laws, at least within reasonable size ranges, and it is plausible that a power law better describes primary production as well. Perhaps the log-normal assumption is justifiable on observational or theoretical grounds (although I am not aware of specific references to support or disprove it), but at least some sensitivity to this assumption should be shown.

Likewise, other model assumptions need stronger support. For example, in marine ecosystem, the prey size range tends to increase with the size of the predator: unicellular organisms are better at feeding on similar-sized microorganisms, while larger predators feed on proportionally smaller sizes. Thus, prey size is not constant with size (e.g. Barnes, 2010, Ecology). Again, the assumption of constant prey size should be accompanied by an analysis showing that ignoring variations does not affect the results of the model. The same could be said about the trophic efficiency, which is likely not constant between the size range of unicellular micro-zooplankton and that of predatory fish.

Framing the implications of the manuscript as relevant for the impacts of current global warming is somewhat of a straw man — as the Authors note, a million-year long record of fish production is not relevant to the decadal to centennial changes experienced by food webs as the ocean ecosystem reacts to anthropogenic climate change. The time scales and processes at play (e.g. evolution) are not comparable. I am not sure any implication can be drawn from the results to the effect of climate change, but if it had to, it would need to be more strongly justified.

Specific comments

Lines 50-52: this reference to picophytoplankton changes in the ocean does not seem relevant to the arguments discussed above, which are mostly about fish.

Lines 63-64: a connection should be made between the proxy of bottom waters and the temperature of the water column at which fish represented by the ichthyolith record (presumably surface or mesopelagic) live. Bottom waters (at least in the modern ocean) reflect high-latitude surface temperatures, so an intermediate step connecting bottom water to global or surface temperature seems necessary.

Lines 86-89. All these sensitivities should be clearly shown, e.g. in a set of figures (in a supplement if space is a constraint).

Lines 95-96: these time series should be shown somewhere as figures.

Lines 109-112: these results are not shown anywhere. The time series of reconstructed primary production, trophic transfer efficiency and other parameters should be shown somewhere.

Lines 115-118: results from these model simulations with positive linear size-primary production relationship should also be shown. Also, the fact that the model is inconsistent with this relationship that is “widely observed” (line 309) gave me pause: it may be interpreted as a lack of consistency of the model with observed relationships, and perhaps should not be brushed under the rug.

Line 119-120: Given the model uncertainties, and the ability of primary production to explain essentially the same amount of information as trophic transfer efficiency, this final interpretation is

somewhat weak. The subtropical gyres are low-productivity regions today — is a 100% increase in production much harder to justify than a 10% change in trophic transfer efficiency? Productivity in the modern ocean varies by orders of magnitude (and subtropical gyres are on the lower end); trophic transfer efficiency should vary perhaps by 10s of percent (given the maximum value of 1), so the two variations may be comparable in relative terms. This should be elaborated on. For example, shifts of physical/biogeographical boundaries in a warmer ocean could cause a significant change in productivity in a region where the absolute productivity is low.

Figure 1: a few aspects should be clarified — the data in panel (a) do not seem to completely match the points in panel (b): in the number of points, which seems to be larger in (a) than in (b), and the minimum maximum temperature in (a) which do not match the minimum and maximum in (b) (e.g. the min T in (a) is >10C, but it is ~9C in (b); the max T in (a) is >15C, but it's <15C in (b)).

Figure 2: the actual units should be provided for Fig. 2, y axis. (is that 1/m or 1/log(m)?)

Reviewer #4:

Remarks to the Author:

This paper uses the geological record to lend insight into assumptions that underpin many projections of the effects of climate change on marine food webs. The authors find that ichthyolith accumulation rate varies in response to temperature, and that the most parsimonious explanation for this variation is that trophic transfer efficiency increases with increasing temperature. These results are important from my perspective as an ecosystem modeler. Results from models are only as good as the models' assumptions. Confronting these assumptions with data is a necessary component of model development. Beyond the ecosystem modeling community, these results will be of interest to those studying climate impacts, marine ecosystems, and marine fisheries.

While I'm unable to evaluate the methods for processing ichthyoliths, the modeling and statistical approaches are valid. The study's conclusions are well supported by the results.

My primary question while reading was about the present speed of warming vs. that during the period of study. The authors address this concern on lines 125 – 129. I also wonder about the consequences of anthropogenic warming resulting in potentially sub-optimal or lethal temperatures. For example, Pörtner and Peck (2010) and Pörtner (2012) suggest that aerobic scope (and presumably trophic transfer efficiency) increase to a point, but above critical temperatures these gains are lost. It would be good for the authors to touch on their implicit assumption that these temperatures weren't reached in their model – and won't be reached during anthropogenic warming. A few minor comments are provided below:

Line 22: "Earth system models consistently predict...." would be more appropriate here.

Line 49: Ref 28 seems to suggest that predation actually decreases at the highest temperatures, challenging the point it's cited here to support.

Line 70: Should this be Ma rather than Myr?

Line 180: There's an errant underscore character between "change" and "thresholds".

Lines 273 – 293: It struck me that there were no references cited in the text describing the trophic model, despite the model following size spectrum modeling conventions. I understand that sometime

work becomes far enough removed from a source to make citation challenging. However, if there are any references that can be pointed to in this section, it would be good to add them in.

Line 300: 50.11 μm seems large for the average size of phytoplankton in a subtropical gyre. I'd expect this to be an order of magnitude smaller (e.g., Polovina and Woodworth 2012).

Fig. 1a: Should the x-axis label be Ma rather than Myr?

Fig 1b: Please add to the legend an explanation of the solid and dashed trend lines.

Fig 2: Should "Myr" used throughout the figure be "Ma"?

Fig. 2: The colors used in this figure could be hard for color-deficient viewers to distinguish, particularly red and green. I recommend revising the color scale so that it is more accessible.

Fig. 2: In the key, the line for primary production should be dashed to match the text in the caption.

For reference since I mentioned them, not to imply that the authors need to cite these papers:

Pörtner, H. O., and Peck, M. A. (2010). Climate change effects on fishes and fisheries: toward a cause-and-effect understanding. *J. Fish Biol.* 77, 1745–1779. doi: 10.1111/j.1095-8649.2010.02783.x

Pörtner, H. O. (2012). Integrating climate-related stressor effects on marine organisms: unifying principles linking molecule to ecosystem-level changes. *Mar. Ecol. Progr. Ser.* 470, 273–290. doi: 10.3354/meps10123

Polovina, J. J., and Woodworth, P. A. (2012). Declines in phytoplankton cell size in the subtropical oceans estimated from satellite remotely-sensed temperature and chlorophyll, 1998 – 2007. *Deep-Sea Res. II* 77-80, 82-88. doi: 10.1016/j.dsr2.2012.04.006

REVIEWER COMMENTS

Reviewer #1 (Remarks to the Author):

This is interestingly controversial manuscript (I am saying positively). Since oligotrophication is predicted with global warming, we tend to think fish production would decrease with warming. The authors found the opposite in paleo record in a much longer time scale. The data and result look robust. It's very important to use paleo-records to understand the present and future ecosystem in proper context, because paleo-records tend to be the only long-term biotic records beyond a few decades. This manuscript does a great job in this sense. The text is well written in general and I enjoyed the reading a lot.

We thank Moriaki Yasuhara for his supportive and constructive review of this manuscript.

A few moderate comments are:

1. Fig 1 says Antarctic Bottom Water Temperature, but this isn't based on global 18O stack, so global deep-water temperature? Sorry if I am wrong.

We thank the reviewer for catching this error, and have fixed it in the figure.

2. It's interesting the fish production has steeper/higher peak at 50 Ma and then rapidly decrease compared to the deep-water temperature (Fig1)

This fish pattern is more similar to SST than deep-water temperature, as the authors can see in Norris et al 2013 Science Fig2 (<https://science.sciencemag.org/content/341/6145/492>). I am not sure if SST data is reasonably available to use for the modeling here. But at least it's worth discussing.

This is an interesting observation – unfortunately, sea surface temperature (SST) records from this region do not exist to compare to, as it is a red clay province, so SST proxies are not preserved. Since SST varies more geographically, and the proxies are less well-accepted and well-calibrated, especially during the extreme warmth of the Eocene, we have chosen to work with common Bottom Water temperature for its more robust, global signal. However, we have added a sentence in the main text to address both this, and the comment below, which are linked. (See discussion on lines 71-75 of tracked changes)

3. Similarly where are fishes from in water column? I mean fish teeth and scales from entire water column deposit on the sea floor eventually. The authors compare the fish result with bottom water temperature that means they are mainly from deep-sea fishes? But fish production must be much higher in ocean surface or shallow marine zone. A bit related to my comment above, so it reflects surface/shallow-water biomass and so makes more sense to compare with SST? Anyway, I think this is an important topic to discuss in the manuscript.

In the modern ocean, the vast majority of fish biomass in pelagic oceans is mesopelagic (Irigoien 2014 Nat Comms). While we have no way of knowing exactly the paleo-vertical distribution of fish biomass in the open ocean in the Eocene, it is most likely that it was similarly mesopelagic-dominated, particularly since many mesopelagic and deep-sea lineages of fishes are more ancient than those that live in epipelagic and coastal environments (see Alfaro 2018 PNAS). Therefore, our use of bottom (“common water”) temperatures is reasonable in this case. In addition, as expressed above, the SST proxies are poorly calibrated and poorly constrained

during this interval, due to the extremely high temperatures, and there is not a SST record for the region that our data are from, and it would be inappropriate to compare directly to SST from a different location in the ocean. Additionally, we wish to note that the model itself does not use temperature as an input, but rather, as a comparison to the output. It is likely that this particular bottom water temperature proxy is more relevant as a general proxy for global climate, rather than being the specific temperature forcing fish metabolic rates. We have added lines 71-75 (tracked changes version) to address this topic.

4. Recent papers showed that pelagic diversity-temperature relationship is unimodal (eg Yasuhara et al 2020 PNAS <https://www.pnas.org/content/early/2020/05/20/1916923117>). Similarly it may be possible that productivity relationship is negative in certain temperature range and positive in in certain temperature range?

Also, Yasuhara et al showed that diversity is highest in the tropical edge/subtropics (ie bimodal latitudinal diversity gradient) in warmer worlds. This study showed biomass is also high there in warmer times. This may mean fishes escaped from too hot tropics to tropical edge/subtropics, resulting high diversity and biomass there in warmer worlds like Eocene and future with RCP 8.5. I think it's interesting to add a discussion on this.

This is an interesting point and excellent paper. We previously noted on lines 109-113 (tracked changes version) that observed morphotype diversity did not change significantly over the warming interval which suggests, citing Sibert et al 2018 Proc B, which examines the diversity from the same record of fish teeth from DSDP Site 596, and demonstrates that there are no significant changes or turnovers in biodiversity during the Early Eocene. Thus, the hypothesis that is put forth here, that fish simply swam away from the tropics and went to the subtropics, is unlikely, as it should be observed as an increase in diversity as tropical species joined the subtropical species, or a turnover as tropical species came in and subtropical species migrated poleward.

Specific comments:

5. It's good that the abstract explain what is "trophic transfer efficiency" a bit more.

Thank you. We have done so on lines 14-15 (tracked changes version).

6. "subtropical" means Paleogene one or present-day one? Or it means more climate regime rather than latitude?

It may be good to better clarify.

We defined subtropical as a latitude band on line 30 (tracked changes version).

7. Line 22 and 24, good to add Mora et al 2013 PlosBiology doi:10.1371/journal.pbio.1001682 to "4,5" and/or "5,7"

We have added the reference.

8. Line 34 "the tiny fraction of the ecosystem that makes it to the seafloor": unclear a bit. rephrase?

We have rephrased this by replacing the "and" with "primarily", to define the fraction we are referring to. We also added reference #17.

9. Line 35, Benthic foraminiferan accumulation rate is widely used proxy for surface primary

production.

The authors may a bit mention/discuss this with citing Yasuhara et al 2012 Paleobiology (<https://doi.org/10.1017/S0094837300000464>); Herguera 2000 MarMic ([https://doi.org/10.1016/S0377-8398\(00\)00041-4](https://doi.org/10.1016/S0377-8398(00)00041-4)); Thomas et al 1995 Paleoceanography (<https://doi.org/10.1029/94PA03056>); etc.

Also, there are organic carbon and calcium carbonate based estimation of primary production widely used.

See eg Yasuhara et al 2009 PNAS (<https://doi.org/10.1073/pnas.0910935106>) and references therein.

This is all excellent information. This group of references is intended to establish that different relationships between have been found between paleo-temperature and paleo-productivity. We have added the Yasuhara et al. 2012 to this effect.

10. Line 61 Subtropical (climate? latitudinal position?) at that time?

Earlier in the manuscript we have defined the subtropical region as approximately 20-40 degrees north/south of the equator.

11. Line 64 Good to briefly mention the method of temperature reconstruction (eg delta 18O based).

Done – thank you!

12. Line 111 etc Myr or myr. Good to be consistent.

Thank you for catching this. Throughout we have chosen to use Myr

13. Line 138-140: The last sentence. The authors may like to broaden the sentence to eg usefulness of microfossil and ocean drilling for ecological and evolutionary research in general beyond the fish with citing recent "biotic response" reviews of Norris et al 2013 Science and Yasuhara et al 2017 Biological Reviews (<https://onlinelibrary.wiley.com/doi/abs/10.1111/brv.12223>). It may help to reach wider audience.

We thank the reviewer for this suggestion. We have edited the sentence to broaden the implications. We declined to mention evolutionary research since our study specifically does not address this aspect of the marine community.

I hope this helps to improve the manuscript.

Please do not hesitate to contact me if there is any unclear point at moriakiyasuhara@gmail.com or yasuhara@hku.hk.

Sincerely yours,
Moriaki Yasuhara

End of the report

Reviewer #2 (Remarks to the Author):

Review of Britten and Sibert for NatComm

Britten and Sibert assess potential causes for observed changes in the size distribution and abundance of fish debris in the Pacific subtropics over the time window encompassing the early Eocene climatic optimum. They note a strong positive correspondence between inferred fish production based on ichthyolith accumulation rate (IAR) and paleotemperature, and model results suggest that changes in trophic efficiency can best explain observed variation in the size distribution over time.

The question of how global climate change affects marine primary production and overall ecosystem production is longstanding and as-yet unresolved. Studies on short-term time scales suggest a negative impact of temperature, while deep-time records hint at the opposite relationship, or at best a complex one. Britten and Sibert use data on fossil fish teeth and debris to estimate changes in production at higher trophic levels and demonstrate a remarkable concurrence between IAR and paleotemperature over the most significant interval of climate warming in the last 65 million years. Figure 1 alone is so surprisingly good that it almost doesn't matter what else the authors say in the text. On a geological time scale, the warmer it gets, the more fish debris there is, in a setting that is otherwise fairly constant in paleoenvironment. This result in itself is enough to warrant publication, as it offers clear solid evidence for higher fish production with warming, contrary to expectations of anthropogenic impacts and pointing to a different set of ecosystem responses to the same perturbation on different time scales.

We thank Dr. Linda Ivany for her supportive and constructive review of our manuscript.

1. With respect to this result, the only thing I'd like to see mention of, and perhaps it was addressed more explicitly in earlier work that I do not have access to at the moment, is what kinds of taphonomic and depositional processes might give rise to variation in IAR, and could these potentially be responsible for the observed pattern.

The reviewer raises a number of interesting points about the fidelity of the ichthyolith record, which we address in turn below. In prior publications (e.g. Sibert et al 2014, Sibert and Norris 2015, Sibert et al 2017, Sibert et al 2018, Sibert et al 2020, as well as Doyle 1979, Doyle 1985, and others), we have discussed at great length the taphonomy and preservation potential of ichthyoliths on the sea floor. As the reviewer points out, being phosphate, these fossils are extremely durable. Teeth preserved on the seafloor, even in the low pH conditions of the deep Pacific red clays, are in excellent condition, with no pitting or other signs of chemical degradation even after 50+ million years. We have added a brief line discussing this in the text, lines 66-69 of tracked changes

For example, do fish teeth typically survive passage through the gut of a predator?

We believe that this is the case. While there have been no formal studies of the tooth composition of fish guts or feces, one author (ES) has done some informal taphonomic experiments, placing fish jaws and teeth into various acidic solutions to mimic fish guts, and it takes weeks to dissolve the teeth, suggesting that teeth do typically survive passage through the gut of a predator.

Could more teeth just mean fewer large predators, so more teeth survive? Given that the shape of the abundance distribution doesn't change, only the height, I don't think this is likely.

We agree with the reviewer here that due to the lack of change in the abundance distribution, it is unlikely that more teeth simply means more smaller fish and fewer big ones.

How long does it take to dissolve them on the sea floor? Being phosphate, I suspect this is not a big issue either.

This is correct – teeth and denticles are some of the oldest vertebrate fossils, dating back over 400 million years. These teeth are from a red clay core, and easily survived >50 million years in acidic conditions that were destructive to all other microfossils. Indeed, there are well-preserved teeth in this same core going back at least 85 million years (Sibert et al 2016), so this is not an issue in this case.

It would be nice though to know something about how IAR scales with production over the modern ocean to be sure that the signal is indeed one related to production and not some other process.

While it is the work of many upcoming years to determine how IAR scales with production in the modern ocean (an ongoing research effort), we point to Sibert et al 2020, which shows IAR from a global array of sediment cores across the Eocene/Oligocene, and note that in this case, IAR scales by the same order of magnitude as modeled modern fish productivity based on several different models from the FishMIP modeling project. While of course the Eocene/Oligocene is not modern boundary conditions, this strong global correlation between IAR and fish productivity, along with other in-prep datasets from the Pliocene, suggest that IAR does, in fact, scale with productivity. However, as this hinges on as-yet un-published comparisons between data and models, we feel it is inappropriate to add to the main text at this point.

Following this, the authors go on to use the size frequency distributions from different time slices to test ecosystem models that explore the influence of various potentially causal processes. If tooth size scales with body size, then those data can be used to infer trophic structure at levels above the primary producers. A food web model then attempts to reproduce the changes seen over time by changing one of several variables to see how well it captures the observed variation. Because the mean and range of the size distribution do not shift, but overall abundance does, efficiency of energy transfer between trophic levels emerges as the most likely driver over time. Overall production could effect the same change but would likely change the size distribution as well, adding a larger tail, and this is not observed. Predator-prey size ratios and/or variability would also change the mean and range of the distribution and so can be ruled out.

2. These inferences are quite interesting because one need not change overall primary production to see a change in production at higher trophic levels. This has been a matter of great debate in the literature regarding the fossil record does primary production change, and if so, how is that change brought about and sustained on a global and/or long-term basis? Perhaps that is not needed at all to explain some of the first-order trends seen in the fossil record. The flip side of this, if primary production is not driving change at higher trophic levels, is how does one change trophic efficiency? This is a discussion that the authors could stand to bolster in their manuscript. Its treatment now is somewhat cursory. Do fish simply eat more of what is available to them?

How and why? The authors admit that the mechanism is unclear, but more discussion than what is present in the paragraph beginning on line 121 is really needed.

In response to other reviewer comments, we have restructured the results slightly to elevate the potential role of primary production in explaining the observed record since we are able to quantitatively explain the record similarly with trophic efficiency and primary productivity. Our extended model analyses lend some additional support for trophic transfer efficiency over primary production; however, we cannot discount primary production changes. Accordingly we have restructured the writing to divide the four models/hypotheses into two classes – one associated with energy flow (trophic transfer efficiency and primary production) and one associated with food web connections and predator-prey relationships (mean size and generalism of prey) – please see discussion on lines 117-123 (tracked changes) We can confidently ascribe the observed changes to mechanisms associated with energy flow while refuting hypotheses related to predator-prey structure. In response to the reviewer, we have also added additional discussion around how temperature may increase predator-prey attack rates and increase the productivity of the microbial loop, thus routing dissolved organics to higher trophic levels (lines 139-146 of tracked changes). There remains significant uncertainty around the temperature effect on food web metabolism – with several hypothesized effects that go in opposite directions.

3. Lastly, I completely agree that time scale is everything when comparing these results to the response to anthropogenic warming. The former deals with a macroevolutionary response, while the latter must be almost exclusively ecological. One wonders then whether these data really do provide any insights to ‘better understand climate impacts on fish communities’ in the context of the present (which is how many will read ‘climate impacts’ when the word ‘anthropogenic’ is also in the same paragraph). I might tweak the language in that last paragraph to be careful not to conflate the two processes and risk being misquoted in the future.

Thank you for the suggestion. We have changed the wording to ‘better understand the role of long term climate on fish community productivity’ in order to broaden the statement (lines 160-163 of tracked changes)

This is an interesting and compelling dataset and a thought-provoking analysis. The issues dealt with are important and longstanding ones. The model results might raise more questions than answers, but nonetheless seem to be able to rule out some proposed mechanisms for change. The text is well written and concise. This should be published.

We thank the reviewer for her supportive and constructive comments.

Some minor editorial comments:

I was a bit confused in Figure 2 by the colored lines the influence of primary production and trophic transfer appear to be exactly the same in each of the model runs. If this is not an error, it should be clarified. I was also unclear how the curve for mean prey size was so different in boxes that otherwise appear quite similar (e.g., 53-52 Myr versus those adjacent).

It is correct that the numerical effect of primary productivity is very similar to trophic transfer efficiency. Variations in both parameters can explain the observations almost exactly as well in terms of the sum of squared error. Mathematically, the differences in the parameters’ effects can

be increased by changing the allometric parameters (causing some divergences at larger size), however our optimized allometric parameters put the curves almost completely on top of one another. As noted above, we have increased the emphasis on primary productivity as a potential explanation for this reason. A visualization of each parameter's effect on the model has also now been added to the supplementary material.

Secondly, the point about large shifts in the mean prey size curves despite similar looking data is well taken. We found that the optimizations for mean prey size and the size-dependent productivity factor results in 'regimes' where the parameters would oscillate between a high and low parameter regimes due to relatively small changes in the data, resulting in very different curves. For example, in time bin 54-53 Ma the green curve stays very close to the optimized reference values, but when the data shift downward in bin 53-52, the optimized parameter drops to a very low value in order to minimize the sum of the squared error between the data curve and the green curve. These regimes highlight the nonlinearity of the optimization problem and point to the fact that very different looking curves can result in similar sum of squared errors. We now show the time-series of optimized time-varying parameters in the supplementary information which demonstrates the regime-like behavior. The unstable fits of mean prey size help demonstrate the inability for changes in mean prey size to explain the data.

4. There are some places where a word appears to be missing in the text:

Line 88 predatory-prey size ratio? smooths

Line 116 factor of three relative TO the time-varying

Line 118 inconsistent with THE ichthyolith record

Line 123-124 overcompensate FOR increases in

Line 274 tooth SIZE distribution?

Line 310 increases in Pu and ARE inconsistent with the ichthyolith data.

Line 316 solid lines give the sample SD - do you mean dashed lines?

Line 323 RMSE computed FOR each Myr bin

Thank you for catching these, they have all been rectified in the text!

Reviewer #3 (Remarks to the Author):

The manuscript by Britten and Sibert discusses a multi-million year record of microfossil fish teeth (ichthyoliths) from the South Pacific subtropical gyre, corresponding to a period of extreme global warming during the Early Paleogene Period (62-46 million years ago), showing that fish production (inferred from the total ichthyoliths accumulation rate) is positively correlated with proxies of ocean temperature. The Authors further use a simple model of the marine food web to argue that this correlation is best explained by changes in the efficiency with which energy (or biomass) is transferred from the base of the food web to fish.

This is a concise paper that merges paleo-ecosystem records and modeling to probe the influence of a fundamental environmental variable, temperature, on a fundamental ecological property fish production. The approach and results are very original and provocative, and if they could hold up to scrutiny they would be very significant for a broad community.

I have no particular reservation on the ichthyoliths record and its correlation with paleo-proxies of temperature, although this is not my field of expertise. In themselves, the paleo-fish results are exciting, and worth of publication. I am more skeptical about the model and ecological interpretation of the ichthyolith record, which form the backbone of the manuscript.

While I am supportive of the idea of combining ecological models and paleo-data, I do not think the analysis convincingly support the interpretation of the paleo-record. Additionally, while I appreciate the conciseness of the manuscript, I feel more could be done to support the results and interpretation with only two figures, very little material is presented to support many of the claims, making it often hard to judge the robustness of the conclusions. Thus, while provocative, I would place low confidence on the idea that higher temperature drives an increase in fish productivity by way of an increase in

trophic transfer efficiency. Given the potentially important implications, this conclusion should be better supported.

We thank the reviewer for the constructive comments. We first wish to point out that the modeling is only to aid the interpretation of the novel empirical ichthyolith accumulation rate data and strong statistical association between ichthyolith accumulation rate and an independent paleo-temperature reconstruction. In our opinion, these data form the backbone of the paper. That said, we agree that the evidence for changes in trophic transfer vs. primary production is not quantitatively conclusive. For this reason we have revised some of the writing to reflect this; in particular, we have conceptually separated the models into those dealing with energy flow (trophic transfer efficiency and primary productivity) and those dealing with predator-prey structural relationships (mean prey size and generalism). We describe our results as supporting changes in energy flow and refuting changes in predator-prey relationships. We now make this point on lines 117-123 (tracked changes). We also note the important point that the trophically aggregated data severely limits the model extensions that can be tested with the data. We discuss this point in more detail below.

General comments

1. I am skeptical of the interpretation of the total ichthyoliths accumulation rate as a straight proxy for fish productivity, as discussed in the methods (line 272), and implied by Figure 1. This is a technical point, but it is relevant to the results, and also relates to the size-dependent model of fish production and its interpretation. The total number of ichthyoliths is the sum of the ichthyoliths across all sizes sampled. Each ichthyolith size corresponds roughly to a fish size, e.g. following the relationship given by the equation after line 275. Individual fish production (e.g. biomass produced per unit time per fish) is also a function of size, presumably proportional to $\text{mass}^{2/3}$ or $\text{mass}^{3/4}$ (see for example Brown's metabolic theory of ecology, Brown et al., 2004, Ecology). Given that a fish's mass is proportional to size^3 , fish production should be proportional to size^2 or $\text{size}^{2.25}$, depending on the allometric relationship used for fish production. Thus, to estimate the total fish production based on a size spectrum of fish numbers (like the one presented by the Authors), one should multiply the number of ichthyoliths in a given size bin to a term proportional to $(\text{ichthyolith size})^{(b*2)}$ or $(\text{ichthyolith size})^{(b*2.25)}$, before summing up (i.e. integrating) over the ichthyolith size range. In a sense, larger ichthyoliths should be weighted more than smaller ones, because they correspond to larger fish with a proportionally larger productivity. Thus, the assertion in the methods (line 272) that "the area under the curve is a proxy for total fish community production" seems incorrect, or at least

incomplete. Given the observations in Fig. 2, it is not clear how the larger weight that should be given to larger fish could compensate for the lower abundance, thus altering the results of the paper, in terms of production by the entire fish community, rather than total number of ichthyoliths accumulated.

We thank the reviewer for the thoughtful comments. Indeed we have thought about and discussed the application of metabolic theory to these data. Firstly we note that length vs. volume relationships vary greatly in fish, with over 50% variation across species in the scaling exponent (e.g. Petrakis and Stergiou 1995, Fisheries Research) which would introduce very large uncertainties into the calculations suggested above. Importantly, we take the area under the IAR size distribution to be a proxy for productivity and not a direct prediction of biomass. We note that total IAR has been well-used in previous papers as a proxy for fish community productivity in deep-time (e.g. Sibert et al 2014, Sibert et al 2020), and generally correlate well with predicted fish biomass in those regions. Finally, there are relatively small numbers of large teeth in these records as compared to small teeth, so although large toothed-fish may disproportionately contribute to biomass, their impact on total biomass is small.

2. I am also worried that the ecological interpretation is strongly dependent on model assumptions that are not clearly laid out or justified, and whose implications are not systematically evaluated and presented to the reader. The model in itself seems simple, but many of its aspects remain unclear, and very few references are provided to justify its formulation. With plenty of space available to present more analyses in the manuscript, the Authors could have strived to show that many of the uncertain model assumption are indeed not relevant for the final conclusions.

We hope the comments and revisions outlined below will address the reviewer's concerns. As it touches on several comments below, we point out here that the modeling is fundamentally limited in its complexity by the available data. The most important limitation is that the data are trophically-aggregated, while we are fitting a trophically-structured model. We sum the model across trophic level when compared with the data. This makes testing trophically-defined model extensions extremely difficult, and in fact statistically non-identifiable in most cases. The model is both simple and flexible and is capable of reproducing a wide-variety of size-distributions. We have kept the assumptions as minimal as possible in order to isolate the effects of the four key parameters. We will refer back to this point in multiple responses below. We now explicitly make this point in the main text. (Lines 96-100 of the track changes version)

3. For example, is the model supposed to represent just the fish community or the entire predator community? At some sizes, fish may overlap with other predators (zooplankton, crustaceans at small sizes, cephalopods at larger sizes), but it is not clear that this is accounted for in the model, at least considering the roughly 50-fold size range shown in Fig. 2. The assumption seems that within the size range implied by the figure, fish are the only important predators; this may be true or not for the Paleogene ocean, but it should be clearly stated and supported.

This is an important point. There is a constant and unknown proportionality that is carried through the model from primary production to fish. This proportionality absorbs both the unknown conversion factor between tooth number and fish biomass; and secondly, the unknown proportion of fish to entire consumer community. We have made the assumption here that the proportion of fish to the rest of the consumer community is approximately constant with trophic level. Unfortunately, this is a case where the trophically-aggregated data are insufficiently

resolved to empirically test these trophically-defined model extensions. We have now expanded on the discussion of this proportionality in the text. (Lines 338-340 of the tracked changes)

4. The main equation of the model (following line 282) asserts that the production rate in one trophic level and size class is a weighted sum over the productions of all preys at the previous trophic level (with a log-normal prey preference weight). I have a hard time convincing myself that this formulation is correct, given that it is quite different from other size-structured food web models that I have seen in the past (e.g. starting with the archetypal size-structured food web model of Benoit and Rochet, 2004, Journal of Theoretical Biology), where the productivity at one trophic level does not come directly from the prey productivities at the previous trophic level, but from the prey biomass at the previous trophic level, multiplied by some grazing rate kernel. It may be that the formulations are equivalent in the end, but a more solid justification for this equations relating productivities to productivities seems required, given how central it is.

We thank the reviewer for this point. The model is a simple size-resolved extension of a trophic pyramid model, where a fraction of productivity at one trophic is transferred to the next. The model directly follows from those in Pauly and Christensen 1995 and Stock et al. 2017. Those papers are now cited in the main text. (Lines 327-328 of the tracked changes)

5. The size spectrum of primary production is assumed to follow a log-normal distribution (given that the size bins are logarithmically spaced), but no justification is provided for this assumption, which in turn probably strongly affects the distribution of the resulting fish productivity spectra in Fig. 2. In general, size spectra in the ocean are better described by power laws, at least within reasonable size ranges, and it is plausible that a power law better describes primary production as well. Perhaps the log-normal assumption is justifiable on observational or theoretical grounds (although I am not aware of specific references to support or disprove it), but at least some sensitivity to this assumption should be shown.

The size distribution of primary productivity and phytoplankton biomass is an interesting topic. We first note that we are assuming productivity distributions with respect to biomass production and not abundance. Log normal biomass distributions have been regularly observed in marine ecosystems whereas power-law distributions in abundance are the norm (e.g. Maranon et al. 2015. Cell size as a key determinant of phytoplankton metabolism and community structure). However, the reviewer is absolutely correct that log power-law biomass distributions are also observed. We experimented with power-law distributions and did not achieve a satisfactory solution. The allometric parameters needed to fit the data were outside of any reasonable range. In the optimization, the power-law distribution in primary productivity can largely be overcome with the appropriate allometric parameters to give a log-normal-looking fish productivity distribution – however the necessary allometric parameters were far outside what we would expect for fish. We now explicitly state this in the paper. (Lines 345-347 of the tracked changes)

6. Likewise, other model assumptions need stronger support. For example, in marine ecosystem, the prey size range tends to increase with the size of the predator: unicellular organisms are better at feeding on similar-sized microorganisms, while larger predators feed on proportionally smaller sizes. Thus, prey size is not constant with size (e.g. Barnes, 2010, Ecology). Again, the assumption of constant prey size should be accompanied by an analysis showing that ignoring variations does not affect the results of the model. The same could be said about the trophic

efficiency, which is likely not constant between the size range of unicellular micro-zooplankton and that of predatory fish.

We thank the reviewer for this point – as it again touches on the key limitations to our model-data synthesis. We unfortunately cannot test trophically-defined model extensions with the available data. The model is extremely flexible so many different shapes of distribution can be achieved. We have limited our analysis to the four key parameters. We have added a sensitivity analysis figure to the supplement demonstrating the impact of changes in model parameters.

7. Framing the implications of the manuscript as relevant for the impacts of current global warming is somewhat of a straw man — as the Authors note, a million-year long record of fish production is not relevant to the decadal to centennial changes experienced by food webs as the ocean ecosystem reacts to anthropogenic climate change. The time scales and processes at play (e.g. evolution) are not comparable. I am not sure any implication can be drawn from the results to the effect of climate change, but if it had to, it would need to be more strongly justified.

This is an important point. In the abstract, introduction, and discussion we have tried to distinguish the impact of warming on short and long timescales. We are providing a novel perspective on long timescales that contrasts from the much more often-discussed anthropogenic timescale. We have now qualified the word ‘warming’ with ‘long-term warming’ throughout the manuscript to make this clearer.

Specific comments

Lines 50-52: this reference to picophytoplankton changes in the ocean does not seem relevant to the arguments discussed above, which are mostly about fish.

We disagree, because picoplankton form the basis of the food web in this ecosystem, so changes in picoplankton will ultimately be passed up to fish.

Lines 63-64: a connection should be made between the proxy of bottom waters and the temperature of the water column at which fish represented by the ichthyolith record (presumably surface or mesopelagic) live. Bottom waters (at least in the modern ocean) reflect high-latitude surface temperatures, so an intermediate step connecting bottom water to global or surface temperature seems necessary.

This is a proxy for planetary climate and is not meant to represent the specific temperature that the fish are experiencing. The fish will live throughout the water column and experience difference temperatures. The deep water temperature proxy is therefore meant to represent an impact of broad variations in climate across depths. We have now pointed that fish are distributed throughout the mesopelagic and discussed their relationship with the proxy on lines 71-75 of tracked changes.

Lines 86-89. All these sensitivities should be clearly shown, e.g. in a set of figures (in a supplement if space is a constraint).

We thank the reviewer for this suggestion which adds a visual component to the explanation of each parameter and the model sensitivity. We have now included this figure as Figure S1 in the supplement.

Lines 95-96: these time series should be shown somewhere as figures.

These time series are displayed as individual distributions in Figure 2. However, we have now also included a time series plot as Figure S2 that shows 10,25,50,75,90 percent quantiles of the observed size distribution with no appreciable trend over time.

Lines 109-112: these results are not shown anywhere. The time series of reconstructed primary production, trophic transfer efficiency and other parameters should be shown somewhere.

We have now added these plots as Figure S3. We thank the reviewer as this inclusion also helped us address additional questions raised in the reviews.

Lines 115-118: results from these model simulations with positive linear size-primary production relationship should also be shown. Also, the fact that the model is inconsistent with this relationship that is “widely observed” (line 309) gave me pause: it may be interpreted as a lack of consistency of the model with observed relationships, and perhaps should not be brushed under the rug.

We now display those results along with the other models in Figure 2.

Line 119-120: Given the model uncertainties, and the ability of primary production to explain essentially the same amount of information as trophic transfer efficiency, this final interpretation is somewhat weak. The subtropical gyres are low-productivity regions today is a 100% increase in production much harder to justify than a 10% change in trophic transfer efficiency?

Productivity in the modern ocean varies by orders of magnitude (and subtropical gyres are on the lower end); trophic transfer efficiency should vary perhaps by 10s of percent (given the maximum value of 1), so the two variations may be comparable in relative terms. This should be elaborated on. For example, shifts of physical/biogeographical boundaries in a warmer ocean could cause a significant change in productivity in a region where the absolute productivity is low.

This point is well taken. We have reworked the writing to better acknowledge primary productivity, as detailed above.

Figure 1: a few aspects should be clarified the data in panel (a) do not seem to completely match the points in panel (b): in the number of points, which seems to be larger in (a) than in (b) *The data in Figure 2b are binned by half Myr time intervals. This is now pointed out in the caption.*

and the minimum maximum temperature in (a) which do not match the minimum and maximum in (b) (e.g. the min T in (a) is >10C, but it is ~9C in (b); the max T in (a) is >15C, but it's <15C in (b)).

Thank you for catching this! There was an error in the axis labeling which has been fixed.

Figure 2: the actual units should be provided for Fig. 2, y axis. (is that 1/m or 1/log(m)?)

Thank you for point this out. Indeed the units are 1/log(m) which has now been added to the axis label.

Reviewer #4 (Remarks to the Author):

This paper uses the geological record to lend insight into assumptions that underpin many projections of the effects of climate change on marine food webs. The authors find that ichthyolith accumulation rate varies in response to temperature, and that the most parsimonious explanation for this variation is that trophic transfer efficiency increases with increasing temperature. These results are important from my perspective as an ecosystem modeler. Results from models are only as good as the models' assumptions. Confronting these assumptions with data is a necessary component of model development. Beyond the ecosystem modeling community, these results will be of interest to those studying climate impacts, marine ecosystems, and marine fisheries.

While I'm unable to evaluate the methods for processing ichthyoliths, the modeling and statistical approaches are valid. The study's conclusions are well supported by the results. *We thank the reviewer for the positive comments, in particular in assessing the validity of the modeling and statistical approaches.*

My primary question while reading was about the present speed of warming vs. that during the period of study. The authors address this concern on lines 125-129. I also wonder about the consequences of anthropogenic warming resulting in potentially sub-optimal or lethal temperatures. For example, Pörtner and Peck (2010) and Pörtner (2012) suggest that aerobic scope (and presumably trophic transfer efficiency) increase to a point, but above critical temperatures these gains are lost. It would be good for the authors to touch on their implicit assumption that these temperatures weren't reached in their model and won't be reached during anthropogenic warming.

This is an excellent point, however on the long timescales addressed in this manuscript, it is unlikely that lethal temperature thresholds would be a concern. In the modern ocean, temperatures are rising rapidly and approaching "lethal" thresholds for organisms adapted to today's cooler ocean temperatures. However, during this interval in Earth's history, the timescale of change was much slower, giving organisms and ecosystems hundreds of thousands (rather than tens to hundreds) of years to adapt to warming conditions. Further, the overall temperature of the planet was much higher during this interval – during the "Paleogene greenhouse", the poles were ice-free, and Antarctica supported a rainforest, and sea-surface temperature proxies regularly record ocean temperatures in excess of 35°C. The organisms and ecosystems were adapted to these warm temperatures, and there is no evidence for extinction or even changes in biodiversity that would be expected had lethal threshold been reached. Throughout the manuscript, we have endeavored to better separate the impacts of short-term, rapid anthropogenic warming, from the long-term processes observed in this study.

A few minor comments are provided below:

Line 22: "Earth system models consistently predict.." would be more appropriate here. *Done. Thank you.*

Line 49: Ref 28 seems to suggest that predation actually decreases at the highest temperatures, challenging the point it's cited here to support. *This is an excellent point. We have altered the language to say that the 'fraction of available prey consumed can increase with warming' so to not imply a monotonic relationship.*

Line 70: Should this be Ma rather than Myr?

Thank you for catching this, it has been fixed in the text

Line 180: There's an errant underscore character between "change" and "thresholds".

Thank you for catching that - the reviewer has quite an eye!

Lines 273-293: It struck me that there were no references cited in the text describing the trophic model, despite the model following size spectrum modeling conventions. I understand that sometime work becomes far enough removed from a source to make citation challenging. However, if there are any references that can be pointed to in this section, it would be good to add them in.

Thank you for raising this. We have now added additional sentences and references for the modeling throughout the methods section.

Line 300: 50 μm seems large for the average size of phytoplankton in a subtropical gyre. I'd expect this to be an order of magnitude smaller (e.g., Polovina and Woodworth 2012).

This is a good point and very true. While we don't have good constraints on the size distribution of phytoplankton in the Eocene, 50 μm is large relative to the modern ocean. We hope the reviewer will understand that inferring the mean phytoplankton size from fish teeth is a very challenging problem! Statistically, the size of the phytoplankton primary productivity distribution strongly trades off against the allometric parameters used to convert between fish body size and fish teeth. In this sense, the phytoplankton size distribution parameters cannot be pinned down precisely, but are better regarded as reference values that can be scaled up or down to examine the impact of variations. We have now added a sentence addressing this point in the methods on lines 351-355.

Fig. 1a: Should the x-axis label be Ma rather than Myr?

Fixed. Thank you.

Fig 1b: Please add to the legend an explanation of the solid and dashed trend lines.

Done. Thank you.

Fig 2: Should "Myr" used throughout the figure be "Ma"?

Fixed. thank you.

Fig. 2: The colors used in this figure could be hard for color-deficient viewers to distinguish, particularly red and green. I recommend revising the color scale so that it is more accessible.

Done. Thank you.

Fig. 2: In the key, the line for primary production should be dashed to match the text in the caption.

Done. Thank you.

For reference since I mentioned them, not to imply that the authors need to cite these papers:

Pörtner, H. O., and Peck, M. A. (2010). Climate change effects on fishes and fisheries: toward a cause-and-effect understanding. *J. Fish Biol.* 77, 1745-1779. doi: 10.1111/j.1095-8649.2010.027833.x

Pörtner, H. O. (2012). Integrating climate-related stressor effects on marine organisms: unifying principles linking molecule to ecosystem-level changes. *Mar. Ecol. Progr. Ser.* 470, 273-290. doi: 10.3354/meps10123

Polovina, J. J., and Woodworth, P. A. (2012). Declines in phytoplankton cell size in the subtropical oceans estimated from satellite remotely-sensed temperature and chlorophyll, 1998-2007. *Deep-Sea Res. II* 77-80, 82-88. doi: 10.1016/j.dsr2.2012.04.006

Reviewers' Comments:

Reviewer #1:

Remarks to the Author:

The MS is revised very well!!

Moriaki Yasuhara

Reviewer #2:

Remarks to the Author:

I'm satisfied with the revisions on this manuscript. Compelling dataset, very interesting and thought-provoking. Not without some controversy, but the best ones usually are. Publish it!

Reviewer #3:

Remarks to the Author:

This is my second review of the paper by Siebert and Britten. In my previous review, I noted that the paper is concise, well written, and presents broadly interesting observations that lead to the surprising conclusion that climatically warm periods support more productive open-ocean food webs, at least over evolutionary-long timescales. I was somewhat more skeptical of the model-supported interpretation that the most likely cause for this temperature-productivity relationship is an increase in trophic transfer efficiency in warmer waters (as opposed to, for example, increases in net primary production).

As the Authors note in the response, the model results are used to support the interpretation of the observations, which by themselves are very compelling. I agree, and I should also note that the idea of using a simple, flexible food web model to interpret this type of paleo-observations is novel and exciting. The Author's reply is convincing in pointing out that this model's simplicity allows to isolate responses to specific processes, which would be harder to constrain or interpret with more complex models. Furthermore, I appreciate the Authors' effort to revise the discussion of the model results to allow for a more nuanced interpretation of potential ecosystem changes. I particularly liked the separation of the model experiments into "energy-flow" and "predator-prey structure" — the resulting discussion feels more balanced. Likewise, other comments by myself and other Reviewers were satisfactorily addressed.

My only remaining concern that has not been satisfactorily addressed in revision relates to my first general comment about the interpretation of the total ichthyoliths accumulation rate as a straight proxy for fish productivity. Now, I don't think this is a fatal flaw of the paper, although I remain convinced that the Authors' interpretation that "the area under the curve is a proxy for total fish community production" is quantitatively incorrect or at least incomplete and should be better justified. I can agree with the Authors that this quantity is a "proxy" for productivity, but I stress that, because of the generally accepted allometric dependence of productivity on size (e.g. Brown's Metabolic Theory of Ecology), there is not a linear correspondence between the area under the curve and actual productivity, and in fact the former is likely to underestimate the importance of larger sizes for productivity and total fish mass (as an example taking biomass as a variable, a single fish of 10cm corresponds to the biomass of 1,000 fish of 1cm, not 10!).

Now, I think the Authors should not brush under the rug this point, as they seem to do in revision. Rather, at the least, they should be forward in acknowledging the limitations of this interpretation, for

example adding a paragraph to the methods section where they discuss assumptions and caveats of interpreting the “area under the curve” as a straight proxy of production. A test where an allometric relationship is applied (e.g. from the MTE, with a typical 3/4 or 2/3 exponent for individual productivity vs. mass) would be more convincing.

With respect to the Reviewers’ answers to my comment, I note that:

“length vs. volume relationships vary greatly in fish, with over 50% variation across species in the scaling exponent (e.g. Petrakis and Stergiou 1995, Fisheries Research) which would introduce very large uncertainties into the calculations suggested above.” This is correct, but the fact remains that the relationship is allometric, and not linear. Adoption of typically accepted values (chiefly, 3/4), perhaps with some form of tuning, would be enough to get a sense of the importance of using allometric relationships to relate fish teeth numbers to production, and the role of small vs. large sizes in the fish size spectrum. This would not be dissimilar to the first assumption of the model, that body size scales allometrically with tooth size (lines 318-319).

“Importantly, we take the area under the IAR size distribution to be a proxy for productivity and not a direct prediction of biomass.” Correct, but this may have important unintended consequences, primarily by giving more weight to smaller sizes rather than larger sizes, in a way that has not been assessed.

“We note that total IAR has been well-used in previous papers as a proxy for fish community productivity in deep-time (e.g. Sibert et al 2014, Sibert et al 2020), and generally correlate well with predicted fish biomass in those regions.” These papers are from the same first Author, so they carry limited weight as to whether the assumption is broadly accepted and ultimately correct or not. The correlation with biomass is to be expected (I agree on the use of total IAR as a proxy for it), but it does not say much about variations driven by size structure and their potential change over the geological record.

“Finally, there are relatively small numbers of large teeth in these records as compared to small teeth, so although large toothed-fish may disproportionately contribute to biomass, their impact on total biomass is small.” This again has not been demonstrated, and the importance or not of few large teeth depends not just on their number, but on the exponent of the allometric relationship used, and the fish teeth size distribution itself. It may be very well that the Authors are correct and in the end larger fish matter less, but this has not been convincingly shown. Also, note that larger teeth, by nature of limited sample size, may also be underestimated in the samples, as they become rarer.

In summary, with relatively few caveats in the manuscript or methods, this point could at least be given adequate consideration, if not fully resolved.

Reviewer #4:

Remarks to the Author:

The authors have addressed nearly all my comments. I particularly appreciate the clearer distinction between the long timescale investigated here and the much more rapid pace of current anthropogenic warming.

The only concern that remains is the color palette in Fig. 2. It is still quite challenging for color-deficient viewers to interpret, and the colors in the figure don’t seem to match those listed in the caption. Options for testing whether colors are universally interpretable include Adobe Illustrator’s

“Proof Setup” view options and the website <https://www.color-blindness.com/coblis-color-blindness-simulator/>. Alternatively, <https://colorbrewer2.org> provides information on creating colorblind-safe palettes.

A few other minor points are:

Line 30: “Approximately 20 – 40° latitude” would be more straightforward.

Line 147: “anthropogenic vs. global timescales” is confusing. “Global” conveys a measure of area rather than of time. Is it possible to use “geological timescales” or something similar?

Figure S1: Please define the four variables plotted, either within the legend for each panel or in the figure caption.

REVIEWERS' COMMENTS

Reviewer #1 (Remarks to the Author):

The MS is revised very well!!

Moriaki Yasuhara

We thank Moriaki Yasuhara for the support and for the constructive comments in the earlier review.

Reviewer #2 (Remarks to the Author):

I'm satisfied with the revisions on this manuscript. Compelling dataset, very interesting and thought-provoking. Not without some controversy, but the best ones usually are. Publish it!
We thank the reviewer for the support and for the constructive comments in the earlier review.

Reviewer #3 (Remarks to the Author):

This is my second review of the paper by Siebert and Britten. In my previous review, I noted that the paper is concise, well written, and presents broadly interesting observations that lead to the surprising conclusion that climatically warm periods support more productive open-ocean food webs, at least over evolutionary-long timescales. I was somewhat more skeptical of the model-supported interpretation that the most likely cause for this temperature-productivity relationship is an increase in trophic transfer efficiency in warmer waters (as opposed to, for example, increases in net primary production).

As the Authors note in the response, the model results are used to support the interpretation of the observations, which by themselves are very compelling. I agree, and I should also note that the idea of using a simple, flexible food web model to interpret this type of paleo-observations is novel and exciting. The Author's reply is convincing in pointing out that this model's simplicity allows to isolate responses to specific processes, which would be harder to constrain or interpret with more complex models. Furthermore, I appreciate the Authors' effort to revise the discussion of the model results to allow for a more nuanced interpretation of potential ecosystem changes. I particularly liked the separation of the model experiments into "energy-flow" and "predator-prey structure" — the resulting discussion feels more balanced. Likewise, other comments by myself and other Reviewers were satisfactorily addressed.

We thank the reviewer for these comments.

My only remaining concern that has not been satisfactorily addressed in revision relates to my first general comment about the interpretation of the total ichthyoliths accumulation rate as a straight proxy for fish productivity. Now, I don't think this is a fatal flaw of the paper, although I remain convinced that the Authors' interpretation that "the area under the curve is a proxy for total fish community production" is quantitatively incorrect or at least incomplete and should be better justified. I can agree with the Authors that this quantity is a "proxy" for productivity, but I stress that, because of the generally accepted allometric dependence of productivity on size

(e.g. Brown's Metabolic Theory of Ecology), there is not a linear correspondence between the area under the curve and actual productivity, and in fact the former is likely to underestimate the importance of larger sizes for productivity and total fish mass (as an example taking biomass as a variable, a single fish of 10cm corresponds to the biomass of 1,000 fish of 1cm, not 10!).

Now, I think the Authors should not brush under the rug this point, as they seem to do in revision. Rather, at the least, they should be forward in acknowledging the limitations of this interpretation, for example adding a paragraph to the methods section where they discuss assumptions and caveats of interpreting the "area under the curve" as a straight proxy of production. A test where an allometric relationship is applied (e.g. from the MTE, with a typical $3/4$ or $2/3$ exponent for individual productivity vs. mass) would be more convincing.

With respect to the Reviewers' answers to my comment, I note that:

"length vs. volume relationships vary greatly in fish, with over 50% variation across species in the scaling exponent (e.g. Petrakis and Stergiou 1995, Fisheries Research) which would introduce very large uncertainties into the calculations suggested above." This is correct, but the fact remains that the relationship is allometric, and not linear. Adoption of typically accepted values (chiefly, $3/4$), perhaps with some form of tuning, would be enough to get a sense of the importance of using allometric relationships to relate fish teeth numbers to production, and the role of small vs. large sizes in the fish size spectrum. This would not be dissimilar to the first assumption of the model, that body size scales allometrically with tooth size (lines 318-319).

"Importantly, we take the area under the IAR size distribution to be a proxy for productivity and not a direct prediction of biomass." Correct, but this may have important unintended consequences, primarily by giving more weight to smaller sizes rather than larger sizes, in a way that has not been assessed.

"We note that total IAR has been well-used in previous papers as a proxy for fish community productivity in deep-time (e.g. Sibert et al 2014, Sibert et al 2020), and generally correlate well with predicted fish biomass in those regions." These papers are from the same first Author, so they carry limited weight as to whether the assumption is broadly accepted and ultimately correct or not. The correlation with biomass is to be expected (I agree on the use of total IAR as a proxy for it), but it does not say much about variations driven by size structure and their potential change over the geological record.

"Finally, there are relatively small numbers of large teeth in these records as compared to small teeth, so although large toothed-fish may disproportionately contribute to biomass, their impact on total biomass is small." This again has not been demonstrated, and the importance or not of few large teeth depends not just on their number, but on the exponent of the allometric relationship used, and the fish teeth size distribution itself. It may be very well that the Authors are correct and in the end larger fish matter less, but this has not been convincingly shown. Also, note that larger teeth, by nature of limited sample size, may also be underestimated in the samples, as they become rarer.

In summary, with relatively few caveats in the manuscript or methods, this point could at least be given adequate consideration, if not fully resolved.

We appreciate the reviewer's detailed comments on whether the ichthyolith accumulation rate (IAR) should be used as a direct proxy for fish productivity. While a nonlinear transformation of the data could in theory be used to calculate productivity, we stress again these calculations would be highly uncertain due to the uncertainty in nonlinear scaling coefficients (as demonstrated in the cited reference - Petrakis and Stergiou 1995). Secondly and most importantly, we show that the ichthyolith size distribution does not change over the analyzed record, so the suggested calculations would simply scale the productivity up and down and not impact the temporal trend or correlation with temperature, which are the topics of the paper. We interpret the sign and strength correlation rather than an estimated dimensional slope. Finally, we strongly disagree that published papers using IAR as a productivity proxy should be discounted because they included one of the authors of this paper. The literature on this topic is small and the author in question has helped lead the field. All previously published papers that use IAR as a productivity proxy were, by virtue of being published, subjected to rigorous peer review.

To directly address these points in the paper, we have now added a sentence directly referencing the papers that use IAR as a direct measure of productivity (lines 326-327 - tracked changes version) and have also added a sentence acknowledging that the estimated allometry could in theory be used to extrapolate total IAR to total biomass, but would be largely inconsequential for the correlations with temperature since there is no trend in the size distribution over time and would introduce unnecessary uncertainty into the calculations (lines 358-360 - tracked changes version).

Reviewer #4 (Remarks to the Author):

The authors have addressed nearly all my comments. I particularly appreciate the clearer distinction between the long timescale investigated here and the much more rapid pace of current anthropogenic warming.

The only concern that remains is the color palette in Fig. 2. It is still quite challenging for color-deficient viewers to interpret, and the colors in the figure don't seem to match those listed in the caption. Options for testing whether colors are universally interpretable include Adobe Illustrator's "Proof Setup" view options and the website <https://www.color-blindness.com/coblis-color-blindness-simulator/>. Alternatively, <https://colorbrewer2.org> provides information on creating colorblind-safe palettes.

The revised Figure 2 was constructed with the 'Dark2' color palette within the 'RColorBrewer' package. It was chosen specifically as a colorblind-friendly palette in response to the reviewer's previous comment. We are happy to modify the color scheme if the editorial office deems the figures colorblind unfriendly.

A few other minor points are:

Line 30: “Approximately 20 – 40° latitude” would be more straightforward.

Done - thank you (line 30 - tracked changes version).

Line 147: “anthropogenic vs. global timescales” is confusing. “Global” conveys a measure of area rather than of time. Is it possible to use “geological timescales” or something similar?

Done - thank you (lines 152-153 of tracked changes version).

Figure S1: Please define the four variables plotted, either within the legend for each panel or in the figure caption.

Done - thank you.